# The association between haemoglobin levels in the first 20 weeks of pregnancy and pregnancy outcomes

Deborah A. Randall[1,2]*, Jillian A. Patterson[1,2], Felicity Gallimore[1,2], Jonathan M. Morris[1,2], Therese M. McGee[3,4], Jane B. Ford[1,2], for the Obstetric Transfusion Steering Group¶

**1** The University of Sydney Northern Clinical School, Women and Babies Research, St Leonards, New South Wales, Australia, **2** Northern Sydney Local Health District, Kolling Institute, New South Wales, Australia, **3** Women's and Newborn Health, Westmead Hospital, Westmead NSW, Australia, **4** The University of Sydney, Sydney NSW, Australia

¶ Membership of the Obstetric Transfusion Steering Group is provided in the Acknowledgments.
* deborah.randall@sydney.edu.au

**Data Availability Statement:** The dataset used in this paper was created by an extraction of maternal clinical data from the ObstetriX database and the Electronic Medical Record at Royal North Shore

## Abstract

### Background

Low haemoglobin has been linked to adverse pregnancy outcomes. Our study aimed to assess the association of haemoglobin (Hb) in the first 20 weeks of pregnancy, and restoration of low Hb levels, with pregnancy outcomes in Australia.

### Methods

Clinical data for singleton pregnancies from two tertiary public hospitals in New South Wales were extracted for 2011–2015. The relationship between the lowest Hb result in the first 20 weeks of pregnancy and adverse outcomes was determined using adjusted Poisson regression. Those with Hb <110 g/L were classified into 'restored' and 'not restored' based on Hb results from 21 weeks onwards, and risk of adverse outcomes explored with adjusted Poisson regression.

### Results

Of 31,906 singleton pregnancies, 4.0% had Hb <110 and 10.2% had ≥140 g/L at ≤20 weeks. Women with low Hb had significantly higher risks of postpartum haemorrhage, transfusion, preterm birth, very low birthweight, and having a baby transferred to higher care or stillbirth. High Hb was also associated with higher risks of preterm, very low birthweight, and transfer to higher care/stillbirth. Transfusion was the only outcome where risk decreased with increasing Hb. Risk of transfusion was significantly lower in the 'restored' group compared with the 'not restored' group (OR 0.39, 95% CI 0.22–0.70), but restoration of Hb did not significantly affect the other outcomes measured.

Hospital and Westmead Hospital for the years 2011 to 2015. Approval to use the datasets was obtained from the Health Research Ethics Committee Northern Sydney Local Health District, and through additional Site Specific Assessment applications for each hospital. Data extraction for the ObstetriX database was undertaken by the data custodians at each hospital and the eMR extraction was undertaken by a contracted third party. De-identified data was securely transferred and was then compiled by the authors. As authors, we are not able to share the datasets provided to us. Procedures for obtaining access to the data can be obtained from the Northern Sydney Local Health District Research Office (ph +61 2 9926 4590, https://www.nslhd.health.nsw.gov.au/AboutUs/Research/Office), and the Westmead Women's Institute for Research and Data Collection (W$^2$IRED; contact Assoc/Prof Seng Chai Chua +61 2 8850 8100, https://www.wslhd.health.nsw.gov.au/WNH/Health-Professionals/Research).

**Funding:** The study was funded by the Australian National Blood Authority Pilot Project funding. The funder had no role in study design, data collection and analysis, decision to publish, or preparation of the manuscript.

**Competing interests:** The authors have declared that no competing interests exist.

## Conclusions

Women with both low and high Hb in the first 20 weeks of pregnancy had higher risks of adverse outcomes than those with normal Hb. Restoring Hb after 20 weeks did not improve most adverse outcome rates but did reduce risk of transfusion.

## Introduction

One in 10 women suffer excessive bleeding after childbirth and 15% of these women will have a red blood cell transfusion.[1] An increasing rate of blood transfusion in the maternity population has been noted and is of concern.[2, 3] Pre-natal and antenatal detection and correction of anaemia may be an effective strategy to reduce the impact of blood loss following birth and improve birth outcomes.[4]

Iron requirements increase during pregnancy, mainly to expand red blood cell mass, fulfil the iron requirements of the fetus, and to compensate for blood loss at delivery.[5] While there is an increase in red blood cell volume during pregnancy, there is a larger increase in plasma volume, and this differential increase results in dilution of haemoglobin (Hb) in the blood during pregnancy.[6] In Australia, there is no agreed normal range for Hb concentration in pregnant women[7] and no existing population health data prospectively collecting Hb levels to examine ranges and their impact on outcomes. The World Health Organization (WHO) considers pregnant women with <110 g/L Hb as anaemic,[8] however, the cut-offs are derived predominantly from developing countries and are not necessarily generalisable.[7] The cut-offs recommended by the Centers for Disease Control and Prevention in the United States were developed using four European studies of healthy pregnant women receiving iron supplements.[9] They determined the following maximum Hb levels for diagnosing anaemia: 110 g/L in the first trimester, 105 g/L in the second trimester and 110 g/L in the third trimester.[9] No studies to date have been based on an Australian pregnant population.

Anaemia at the birth admission has been associated with higher caesarean section rates and adverse outcomes such as higher rates of postpartum haemorrhage (PPH), blood transfusion, and infant transfer to neonatal intensive care.[10] Anaemia in the first and second trimester of pregnancy has been associated with low birthweight and preterm birth,[11, 12] as have high Hb levels.[13]

The National Pregnancy Care Guidelines recommend all women have their Hb level checked at the first antenatal visit and again at approximately 28 weeks' gestation.[14] While it is recommended that any anaemia be investigated and treated, routine iron supplementation is not recommended for every pregnancy.[15] Patient Blood Management guidelines aim to reduce obstetric blood transfusions by taking an individualised approach that attempts to reduce the need for transfusion and therefore avoid unnecessary exposure to blood and blood products.[16] Given the potential importance of pre-delivery iron status in helping women cope with blood loss associated with childbirth, it is important to understand the impact of low Hb early in pregnancy on adverse outcomes. At this point, it may be possible to restore the Hb levels. Our study therefore aimed to assess levels of Hb at ≤20 weeks and associations with PPH and blood transfusion at birth or postnatally, and whether restoration of Hb levels reduced the likelihood of PPH or transfusion and/or improved pregnancy outcomes.

## Materials and methods

### Design and setting

The study was a retrospective cohort study using hospital data, and was conducted in two large tertiary public hospitals in New South Wales, Australia, the Royal North Shore Hospital and

Westmead Hospital. Singleton pregnancies in a five-year period from 1 January 2011 to 31 December 2015 were included.

## Data sources

Detailed data on maternal characteristics, pregnancy history, and birth factors were obtained from the ObstetriX database ('birth data'), a clinical database that is completed by midwives at booking, at antenatal visits and at the birth admission, recording information for births of at least 20 weeks' gestation or 400g birth weight. Data on age of the mother, country of birth (COB), inpatient and outpatient encounters for the perinatal period (from the start of pregnancy to six weeks postnatally) for the mother were obtained from the Electronic Medical Record ('eMR'). Most diagnoses and procedures were coded according to the International Statistical Classification of Diseases and Related Health Problems, Australian Modification (ICD10-AM), and the Australian Classification of Health Interventions,[17] with a small minority coded using the SNOMED classification.[18] Hb results and test date were obtained from either the ObstetriX database, where they were manually entered, or eMR Pathology data (where results were from an onsite hospital pathology laboratory, or an off-site linked laboratory for Westmead only).

## Variables

To identify women with low Hb levels, all Hb results obtained in the first 20 weeks of the pregnancy were searched, and the lowest Hb result was classified into 10 g/L categories (<90, 90–99, 100–109 etc to 150–159 g/L), as well as three broad categories of <110 (range 45–109), 110–139 and 140+ (range 140–159) g/L. Hb results between 8 and 14 (n = 24) were re-coded to 80–140 due to likely transcription errors. All other results outside of probable values (45 to 159 g/L), as determined by the clinical authors and comparisons with haematocrit levels, were set to missing. We chose the lowest Hb in the first 20 weeks as we were interested in the Hb before any treatment. Those without a Hb result in the first 20 weeks were excluded from the study, and the distribution of their characteristics compared against the distribution of those in the study using standardized percentage differences.[19] Women with a Hb result <110g/L at ≤20 weeks were further examined to determine their subsequent mean Hb across the remainder of the gestational period before birth (>20 weeks). These women were classified as 'restored' if their mean Hb after 20 weeks gestation was 110 g/L or more, 'not restored' if the mean Hb results were <110 g/L, and 'no further results' if no further Hb results were recorded after 20 weeks and before birth.

Main outcomes were postpartum haemorrhage (PPH; a combined indicator using birth and eMR data), and transfusion at the birth or 6 weeks postnatally (birth and eMR data). Further outcomes included preterm birth (<37 weeks gestation; birth data) divided into planned (induction or pre-labour caesarean) and spontaneous, stillbirth (birth data), neonatal transfer to special care nursery (SCN) or neonatal intensive care unit (NICU; birth data), small for gestational age <10% (SGA; birth data),[20] and very low birthweight (<1500g; birth data). See S1 Table for more details.

Maternal age, COB, and postcode of residence were obtained from the eMR. Parity, smoking, body mass index (BMI), gestational diabetes, pregnancy hypertension, abnormal placenta site, and previous history of anaemia, diabetes, hypertension and major uterine surgery, were obtained from the birth data. Socio-economic status was matched to the Index of Relative Socio-Economic Advantage and Disadvantage (IRSAD)[21] using postcode and divided into population quintiles. Additional birth factors that could be mediating causes of PPH or transfusion, i.e. labour onset, mode of birth, and perineal tears, were obtained from the birth data.

## Statistical methods

Characteristics of women in the three broad Hb groups (<110, 110–139, 140+ g/L) were compared using $\chi^2$ tests. Outcome data were also compared between these broad Hb groups, using percentages. The relative rates of PPH, transfusion, preterm birth, small for gestational age, very low birthweight and transfer to NICU/SCN or stillbirth, were investigated by more detailed Hb groups (<90, 90–99, 100–109, 110–119, 120–129, 130–139, 140–149, 150–159 g/L) using modified Poisson regression models, adjusting for maternal characteristics (age, BMI, country of birth, parity, SES quintile), pregnancy risk factors (smoking, gestational diabetes and hypertension, pre-existing diabetes and hypertension, previous uterine surgery, abnormal placenta site, antenatal haemorrhage), and potentially mediating factors (labour onset, mode of birth, perineal tears). The composite indicator of stillbirth or transfer to higher care was used, as there were not enough stillbirths to look at separately. The risk of these same outcomes for women with their Hb 'restored' and 'not restored' were also estimated using modified Poisson, and adjusted with a smaller set of covariates that differed across the Hb groups and also influenced the outcomes (country of birth, SES quintile, parity, smoking, BMI, labour onset, mode of birth, perineal tears). There were not enough stillbirths or babies with very low birth weights to examine separately, so these were combined in a composite neonatal adverse outcome indicator with transfer to higher care.

## Ethics

Ethical approval for this study was obtained from Northern Sydney Local Health District Ethics Committee (LNR/17/HAWKE/32).

## Results

There were 40,352 births at the Royal North Shore and Westmead hospitals between 1 January 2011 and 31 December 2015. After exclusions (see Fig 1), 31,906 singletons births remained. A number of women were removed from the analysis due to missing Hb in the first 20 weeks (n = 4621). A comparison of characteristics for those women missing Hb and the final analysis population is in S2 Table. Women missing a Hb result in the first 20 weeks of pregnancy were similar to those in the study population, but were less likely to have been born in Southern Asia, and more likely to have been born in Oceania, had 2 or more previous pregnancies, had a previous history of B12/folate deficiency, and have smoked during pregnancy (standardised difference >0.1).

The mean of the lowest Hb at ≤20 weeks recorded for each woman was 127.4 g/L (standard deviation 10.0) and the median was 128 (interquartile range 13). Overall, 4.0% of women had Hb <110 g/L, 85.7% had Hb 110–139 g/L, and 10.2% had Hb 140+ g/L at ≤20 weeks. Women with a Hb <110 g/L at ≤20 weeks were less likely than those with Hb 110–139 and 140+ to be born in Australia, Europe, South- and North-East Asia and the Americas, and were more likely to be born in Africa and the Middle East and Southern and Central Asia (Table 1). There was also a socio-economic gradient, with the likelihood of living in the most advantaged areas increasing with increasing Hb levels. Women with lower Hb had higher parity, and were more likely to have a BMI of <18.5 and have a history of iron-deficiency anaemia, than those with medium Hb, who in turn had higher rates than those with Hb 140+. The low Hb group were more likely to have planned caesareans, less likely to have unassisted vaginal births, and among vaginal births, there were similar rates of perineal trauma.

The most common adverse pregnancy outcomes were infant transferred to higher care (16.0%), PPH (13.6%), a small for gestational age infant (9.5%), and preterm birth (7.1%) (Table 2). Those with Hb <110 g/L had a higher percentage of all adverse outcomes, compared

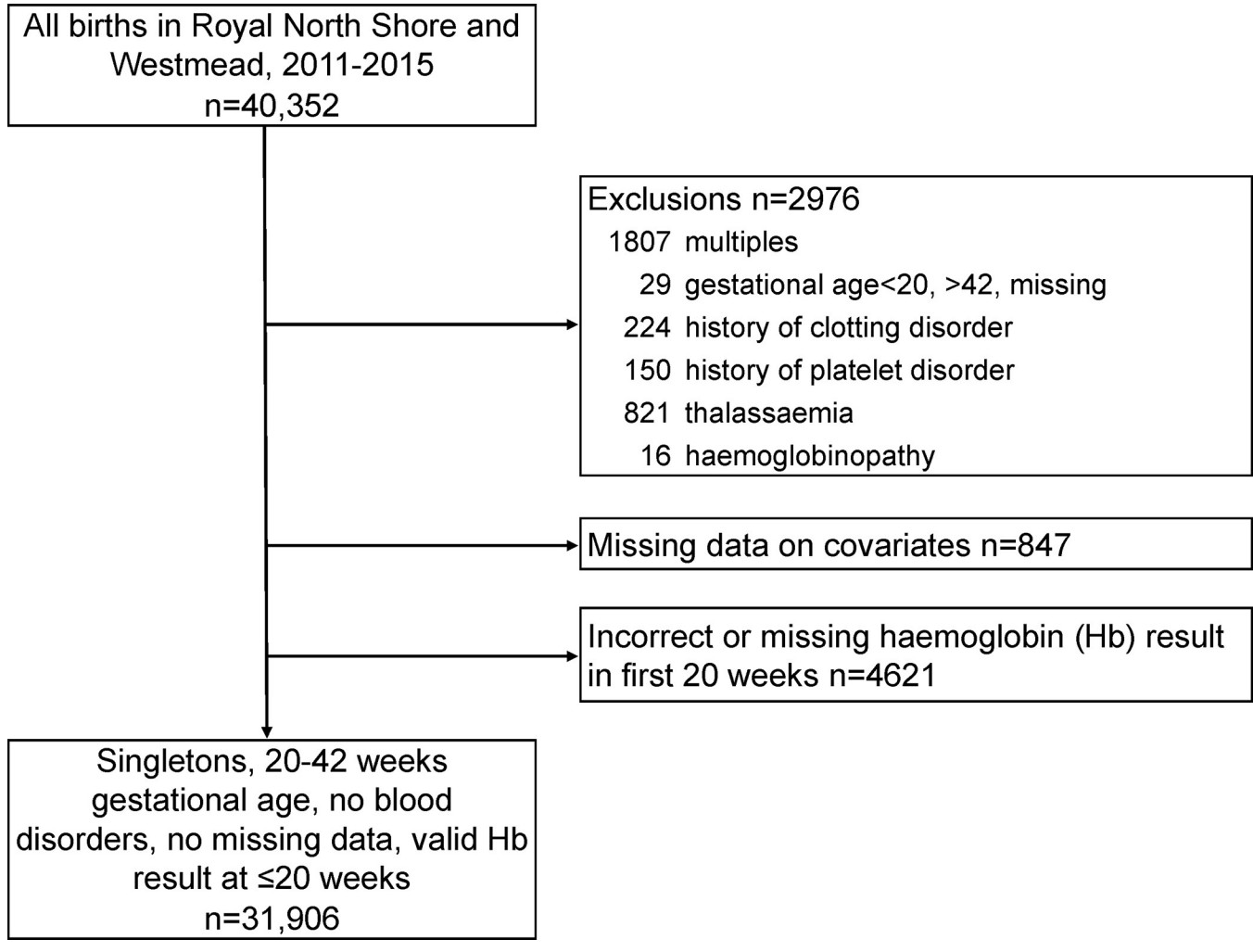

**Fig 1. Study population.**

to the group with Hb 110–139 g/L. Those with Hb 140+ g/L were slightly more likely to have many of the adverse outcomes than the normal Hb group, but were less likely than those with lower Hb to have a transfusion.

Fig 2 shows the adjusted rate ratios for adverse outcomes by the detailed Hb groups. There was a U-shaped relationship between Hb and all of the adverse outcomes except for transfusion, after adjusting for covariates in multivariable models. For most outcomes, the risk was comparatively higher at the lower Hb levels than the higher ones, with the risk starting to increase at Hb levels less than 120 (compared with 120–129 g/L). The U-shaped relationship was only very slight for PPH, with the adjusted rates among those with the highest Hb (150–159) only 1.1 and not significant. There was a linear relationship between Hb and transfusion rate, with the transfusion rate significantly higher in the Hb groups <120 g/L, and lower in the Hb groups ≥130 g/L (although not reaching significance) compared with 120–129 g/L group.

Of 1282 women with Hb <110 g/L at ≤20 weeks, 38% (n = 492) had a mean Hb from 21 weeks to birth of ≥110g/L ('restored'), 38% (n = 488) did not ('not restored'), and an additional 24% (n = 302) did not have any Hb results from 21 weeks onwards (Table 3). We compared the risk of adverse outcomes by 'restored' and 'not restored' Hb groups, adjusting for

**Table 1. Characteristics of the population by hospital of birth and haemoglobin (Hb) cut-points.**

| | Total | | Lowest Hb in first 20 wks | | | | |
| --- | --- | --- | --- | --- | --- | --- | --- |
| | | | <110 g/L | | 110–139 g/L | | 140+ g/L | | χ2 |
| | n = 31,906 | | n = 1282 | | n = 27,356 | | n = 3268 | | test |
| | Col % (n) | | Col % (n) | | Col % (n) | | Col % (n) | | |
| *Maternal characteristic* | | | | | | | | | |
| **Age group‡** | | | | | | | | | |
| <20 | 1.1% | (355) | 1.6% | (21) | 1.1% | (307) | 0.8% | (27) | |
| 20–34 | 76.4% | (24362) | 78.0% | (1000) | 76.2% | (20847) | 77.0% | (2515) | |
| 35+ | 22.5% | (7189) | 20.4% | (261) | 22.7% | (6202) | 22.2% | (726) | |
| **Country of birth‡** | | | | | | | | | |
| Australia | 35.2% | (11227) | 25.0% | (321) | 34.6% | (9457) | 44.3% | (1449) | *** |
| Oceania (rest) | 3.6% | (1134) | 3.7% | (48) | 3.5% | (969) | 3.6% | (117) | |
| Europe | 6.8% | (2158) | 3.5% | (45) | 6.8% | (1867) | 7.5% | (246) | |
| Africa and the Middle East | 10.2% | (3253) | 14.4% | (184) | 10.4% | (2850) | 6.7% | (219) | |
| South- and North-East Asia | 19.8% | (6322) | 13.9% | (178) | 19.7% | (5391) | 23.0% | (753) | |
| Southern Asia | 19.8% | (6332) | 33.6% | (431) | 20.3% | (5543) | 11.0% | (358) | |
| Central Asia | 2.3% | (721) | 4.6% | (59) | 2.3% | (630) | 1.0% | (32) | |
| Americas | 2.4% | (759) | 1.2% | (16) | 2.4% | (649) | 2.9% | (94) | |
| **Socio-economic status quintile‡** | | | | | | | | | |
| 1—most disadvantaged | 21.3% | (6811) | 29.3% | (375) | 21.4% | (5851) | 17.9% | (585) | *** |
| 2 | 4.4% | (1406) | 5.8% | (74) | 4.3% | (1184) | 4.5% | (148) | |
| 3 | 22.6% | (7206) | 29.3% | (376) | 22.6% | (6176) | 20.0% | (654) | |
| 4 | 19.3% | (6147) | 14.8% | (190) | 19.3% | (5276) | 20.8% | (681) | |
| 5—most advantaged | 32.4% | (10336) | 20.8% | (267) | 32.4% | (8869) | 36.7% | (1200) | |
| *Previous history* | | | | | | | | | |
| **Parity†** | | | | | | | | | |
| No previous births | 47.4% | (15117) | 45.7% | (586) | 46.5% | (12713) | 55.6% | (1818) | *** |
| 1 previous birth | 34.5% | (10999) | 31.5% | (404) | 35.0% | (9581) | 31.0% | (1014) | |
| 2+ previous births | 18.1% | (5790) | 22.8% | (292) | 18.5% | (5062) | 13.3% | (436) | |
| **Medical history** | | | | | | | | | |
| Pre-existing diabetes† | 0.9% | (287) | 2.0% | (25) | 0.8% | (231) | 0.9% | (31) | *** |
| Previous gestational diabetes† | 4.3% | (1358) | 4.1% | (53) | 4.3% | (1170) | 4.1% | (135) | |
| Pre-existing hypertension | 6.3% | (1997) | 7.6% | (98) | 6.0% | (1631) | 8.2% | (268) | *** |
| Previous major uterine surgery† | 16.4% | (5229) | 20.1% | (258) | 16.5% | (4526) | 13.6% | (445) | *** |
| **Previous history of anaemia†** | | | | | | | | | |
| Iron-deficiency | 14.9% | (4764) | 44.4% | (569) | 14.3% | (3921) | 8.4% | (274) | *** |
| B12/folate deficiency | 0.3% | (97) | 1.1% | (14) | 0.3% | (74) | 0.3% | (9) | *** |
| *Current pregnancy risks* | | | | | | | | | |
| Smoked during pregnancy‡ | 2.8% | (892) | 2.9% | (37) | 2.6% | (715) | 4.3% | (140) | *** |
| Gestational diabetes‡ | 11.6% | (3688) | 10.9% | (140) | 11.4% | (3107) | 13.5% | (441) | ** |
| Pregnancy hypertension‡ | 5.0% | (1592) | 5.1% | (66) | 4.8% | (1306) | 6.7% | (220) | *** |
| Abnormal placenta site‡ | 1.4% | (437) | 2.2% | (28) | 1.3% | (366) | 1.3% | (43) | * |
| Antepartum haemorrhage‡ | 3.9% | (1260) | 5.3% | (68) | 3.8% | (1044) | 4.5% | (148) | ** |
| **Body Mass Index†** | | | | | | | | | |
| <18.5 | 5.9% | (1879) | 10.0% | (128) | 5.8% | (1589) | 5.0% | (162) | *** |
| 18.5–24.99 | 59.6% | (19023) | 57.6% | (739) | 60.0% | (16403) | 57.6% | (1881) | |
| 25+ | 34.5% | (11004) | 32.4% | (415) | 34.2% | (9364) | 37.5% | (1225) | |
| *Birth factors* | | | | | | | | | |

(*Continued*)

**Table 1.** (Continued)

| | Total | <110 g/L | 110–139 g/L | 140+ g/L | χ2 |
|---|---|---|---|---|---|
| | | | Lowest Hb in first 20 wks | | |
| | n = 31,906 | n = 1282 | n = 27,356 | n = 3268 | test |
| | Col % (n) | Col % (n) | Col % (n) | Col % (n) | |
| **Labour onset** | | | | | |
| Spontaneous | 52.5% (16739) | 47.7% (612) | 52.6% (14381) | 53.4% (1746) | *** |
| Induction | 30.6% (9766) | 30.8% (395) | 30.5% (8338) | 31.6% (1033) | |
| Pre-labour caesarean | 16.9% (5401) | 21.5% (275) | 17.0% (4637) | 15.0% (489) | |
| **Mode of birth** | | | | | |
| Vaginal unassisted | 57.2% (18260) | 52.6% (674) | 57.5% (15736) | 56.6% (1850) | ** |
| Vaginal instrumental–forceps | 7.8% (2480) | 7.3% (93) | 7.8% (2133) | 7.8% (254) | |
| Vaginal instrumental–vacuum | 4.9% (1571) | 5.9% (76) | 4.8% (1323) | 5.3% (172) | |
| Caesarean section | 30.1% (9595) | 34.2% (439) | 29.8% (8164) | 30.4% (992) | |
| **Perineal trauma (for vaginal births only n = 22,311)** | | | | | |
| None | 29.1% (6482) | 33.3% (281) | 28.8% (5536) | 29.2% (665) | |
| 1st degree / Other | 30.3% (6758) | 26.8% (226) | 30.5% (5846) | 30.1% (686) | |
| 2nd degree | 36.1% (8052) | 35.0% (295) | 36.1% (6931) | 36.3% (826) | |
| 3rd or 4th degree | 4.6% (1019) | 4.9% (41) | 4.6% (879) | 4.3% (99) | |

† Recorded at booking

‡ Recorded at birth admission

$\chi^2$ test significance

*$p < 0.05$

**$p < 0.01$

***$p < 0.001$

**Table 2. Outcomes by lowest haemoglobin (Hb) at 20 weeks gestation or less.**

| | Total | <110 g/L | 110–139 g/L | 140+ g/L | χ2 |
|---|---|---|---|---|---|
| | | | Lowest Hb in first 20 wks | | |
| | n = 31,906 | n = 1282 | n = 27,356 | n = 3268 | test |
| | Col % (n) | Col % (n) | Col % (n) | Col % (n) | |
| Adverse maternal outcomes | | | | | |
| Postpartum haemorrhage | 13.6% (4355) | 15.5% (199) | 13.4% (3674) | 14.7% (482) | * |
| Transfusion | 2.1% (659) | 4.9% (63) | 2.0% (542) | 1.7% (54) | *** |
| Preterm birth (<37 wks) | 7.1% (2281) | 10.8% (138) | 6.8% (1848) | 9.0% (295) | *** |
| Planned preterm | 3.7% (1165) | 6.7% (86) | 3.4% (936) | 4.4% (143) | |
| Spontaneous preterm | 3.5% (1116) | 4.1% (52) | 3.3% (912) | 4.7% (152) | |
| Adverse neonatal outcomes | | | | | |
| Stillbirth | 0.7% (235) | 2.0% (25) | 0.7% (186) | 0.7% (24) | *** |
| Small for gestational age | 9.5% (3031) | 12.1% (155) | 9.4% (2582) | 9.0% (294) | ** |
| Very low birthweight (<1500g) | 2.0% (638) | 3.7% (47) | 1.9% (512) | 2.4% (79) | *** |
| Transfer to NICU/SCN | 16.0% (5112) | 19.7% (253) | 15.6% (4272) | 18.0% (587) | *** |

$\chi^2$ test significance:

*$p < 0.05$

**$p < 0.01$

***$p < 0.001$

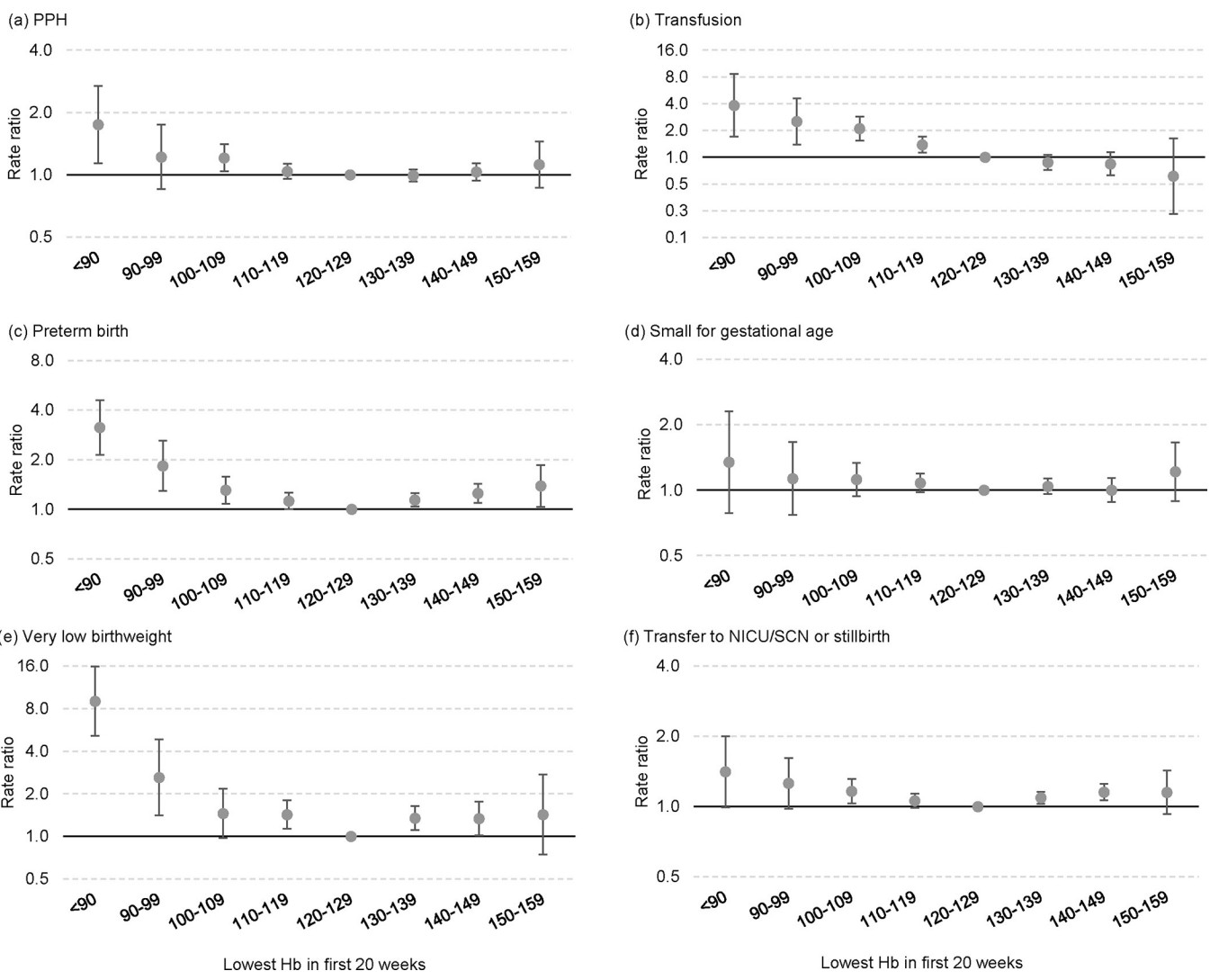

**Fig 2. Adjusted rate ratios for individual outcomes by lowest haemoglobin (Hb) at 20 weeks gestation or less.**

**Table 3. Outcomes among those with low Hb at 20 weeks gestation or less, by Hb status in the remaining weeks.**

| | Total | Hb status in remaining pregnancy | | | | | Restored versus not restored |
|---|---|---|---|---|---|---|---|
| | | No further Hb | | Restored | | Not restored | |
| | n = 1282 | n = 281 | | n = 481 | | n = 472 | |
| | Col % (n) | Col % (n) | | Col % (n) | | Col % (n) | aRR (95% CI) |
| **Adverse maternal outcomes** | | | | | | | |
| Postpartum haemorrhage | 15.5% (199) | 15.6% | (47) | 15.2% | (75) | 15.8% (77) | 0.96 (0.71–1.29) |
| Transfusion | 4.9% (63) | 4.6% | (14) | 2.6% | (13) | 7.4% (36) | **0.39 (0.22–0.70)** |
| Preterm birth (<37 wks) | 10.8% (138) | 11.9% | (36) | 9.1% | (45) | 11.7% (57) | 0.91 (0.63–1.30) |
| **Adverse neonatal outcomes** | | | | | | | |
| Small for gestational age | 12.1% (155) | 11.9% | (36) | 13.6% | (67) | 10.7% (52) | 1.07 (0.75–1.52) |
| Transfer to NICU/SCN or stillbirth or very low birthweight | 21.8% (280) | 25.8% | (78) | 18.7% | (92) | 22.5% (110) | 0.89 (0.70–1.14) |

covariates. The adjusted risk of transfusion was significantly lower in the 'restored' group compared with the 'not restored' group, but there was no significant increased or decreased risk for the other outcomes.

## Discussion

Among 31,906 singleton pregnancies, 4% of women had Hb <110 g/L and 10% had Hb 140 + g/L at ≤20 weeks of pregnancy. Our results suggest that both low and high Hb at ≤20 weeks are associated with adverse outcomes at the time of birth, in a U-shaped relationship that rises on either side of the lowest risk point at 120–129 g/L. The association between the low Hb and adverse outcomes was relatively stronger than that between high Hb and adverse outcomes. Only transfusion had a linear relationship, with risk increasing with lower Hb and decreasing with higher Hb. The U-shaped relationship between Hb and adverse outcomes that we found has also been shown in a study in Peru in both high and low altitude pregnancies. [22]

Of the women with a low Hb at ≤20 weeks, almost 40% had their Hb restored in the second half of pregnancy. Restoration of Hb did not appear to change risk of PPH, preterm birth, SGA or a composite indicator including transfer to higher care, stillbirth and very low birthweight, but did lower the risk of postpartum transfusion. These data are consistent with a review of trial data suggesting iron supplementation improved Hb levels in pregnant women but did not conclusively improve pregnancy outcomes.[23] The reasons why improvements in Hb do not translate into improved perinatal outcomes require further study. There may be a critical window for the impact of low Hb on outcomes, or the low Hb may be a symptom of an underlying condition that is itself the cause of the poor outcome. Another possibility is that restoring Hb does in fact improve some outcomes, but not those specifically measured in our study.

The higher rate of PPH demonstrated in the low compared with normal Hb groups is in line with previous evidence suggesting anaemia is associated with a higher risk of PPH.[24, 25] Women with high antenatal Hb also had a slightly higher PPH rate than those with normal Hb (although not significantly so), but were less likely to be transfused than those with low Hb, which may have been due to better iron reserves or the treating clinicians being more willing to tolerate blood loss before deciding to transfuse. We also found a significantly higher risk of adverse outcomes such as preterm birth, very low birthweight and transfer to higher care or stillbirth for those with high Hb result, as has been found in other studies,[26, 27] possibly due to inadequate plasma volume expansion, or the impaired response to inflammation and infection,[5, 26] or possibly due to high Hb levels before pregnancy.

Antepartum haemorrhage and abnormal placenta site can cause anaemia and are also associated with adverse pregnancy outcomes.[28, 29] These factors were adjusted for in our analysis, but were also unlikely to have influenced anaemia in the first 20 weeks of pregnancy, as bleeding due to these factors usually occurs later in the pregnancy.

Australian data on the prevalence of anaemia in pregnancy is limited. Our estimate of low Hb (4%) was similar to a 2015 South Australian estimate of women with anaemia in pregnancy (6.6%).[30] International studies have found much higher rates of maternal anaemia with a global estimate of 38% in 2011.[31] The high proportion of women with a history of iron-deficiency anaemia in our population (15%), particularly in the low Hb group, suggest there may have been opportunities to correct low Hb due to iron deficiency before the pregnancy.

We were able to obtain Hb results for a large cohort of pregnant women and examine outcomes by Hb levels at ≤20 weeks gestation. However, limitations of these data were that only Hb results that were manually entered in birth data by midwives or were obtained from in-hospital pathology laboratories (at Royal North Shore) or in-hospital or linked pathology

laboratories (at Westmead) were available. This meant there were 13% of pregnant women (n = 4621) who did not have a valid Hb result in the first 20 weeks of pregnancy. These women were broadly similar to those in the final study population, though. Also, we did not know the cause of the low Hb or what measures were taken to restore Hb, and could only infer treatment based on changes in Hb results. From a previous survey, and clinical experience, we assume that a majority of women were taking supplemental iron, either as part of a multivitamin or in an iron-only supplement,[32] but without information on which supplements, and how much iron they contained, collected in the database, we were unable to examine how this impacted on Hb or outcomes. The country of birth results suggest that some thalassaemia/sickle cell anaemia cases may have been missed, as these conditions are more common in Africa and the Middle East, where the low Hb women were more likely to be born.

## Conclusions

We found that women with low Hb in the first 20 weeks of their pregnancy were more likely to have a PPH and blood transfusion after the birth than those with Hb of 120–129 g/L. Those with high and low Hb had increased risk of preterm birth, very low birthweight and a composite outcomes of transfer to higher care or stillbirth compared with those at 120–129 g/L. Restoring the Hb during the pregnancy decreased risk of postpartum transfusion, but did not appear to impact the risk of other adverse outcomes measured. Further research is needed to determine why low and high Hb in the first 20 weeks is associated with poorer outcomes and whether these poor outcomes can be prevented before or during the pregnancy.

## Supporting information

**S1 Table. Detailed codes.**
(DOCX)

**S2 Table. Comparison between study population and those with missing Hb in first 20 weeks.**
(DOCX)

## Acknowledgments

Members of the Obstetric Transfusion Steering Group include: Jennifer R Bowen (Royal North Shore Hospital), Sandra Cochrane (National Blood Authority), David O Irving (Australian Red Cross Blood Service), James P Isbister (University of Sydney), Eleni Mayson (St Vincents Hospital), Michael C Nicholl (NSW Health), Michael J Peek (The Australian National University), Amanda Thomson (Australian Red Cross Blood Service) and Penny O'Beid (Clinical Excellence Commission).

## Author Contributions

**Conceptualization:** Jillian A. Patterson, Jonathan M. Morris, Jane B. Ford.

**Data curation:** Deborah A. Randall.

**Formal analysis:** Deborah A. Randall.

**Funding acquisition:** Jillian A. Patterson, Jonathan M. Morris, Jane B. Ford.

**Investigation:** Deborah A. Randall, Jillian A. Patterson, Felicity Gallimore, Therese M. McGee, Jane B. Ford.

**Methodology:** Deborah A. Randall, Jillian A. Patterson, Felicity Gallimore, Therese M. McGee, Jane B. Ford.

**Project administration:** Jane B. Ford.

**Supervision:** Jonathan M. Morris, Jane B. Ford.

**Writing – original draft:** Deborah A. Randall.

**Writing – review & editing:** Deborah A. Randall, Jillian A. Patterson, Felicity Gallimore, Jonathan M. Morris, Therese M. McGee, Jane B. Ford.

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
