## [Decision Letter · Decision Letter 0]

10 Oct 2019

PONE-D-19-19488

The association between haemoglobin levels in the first 20 weeks of pregnancy and pregnancy outcomes

PLOS ONE

Dear DR Deborah Anne Randall

Thank you for submitting your manuscript to PLOS ONE. After careful consideration, we feel that it has merit but does not fully meet PLOS ONE’s publication criteria as it currently stands. Therefore, we invite you to submit a revised version of the manuscript that addresses the points raised during the review process.

We would appreciate receiving your revised manuscript by November 20th. To enhance the reproducibility of your results, we recommend that if applicable you deposit your laboratory protocols in protocols.io, where a protocol can be assigned its own identifier (DOI) such that it can be cited independently in the future. For instructions see: http://journals.plos.org/plosone/s/submission-guidelines#loc-laboratory-protocols

We look forward to receiving your revised manuscript.

Kind regards,

Massimo Ciccozzi

Academic Editor

PLOS ONE

Journal Requirements:

Reviewers' comments:

Reviewer's Responses to Questions

**Comments to the Author**

1. Is the manuscript technically sound, and do the data support the conclusions?

Reviewer #1: Yes

Reviewer #2: Yes

2. Has the statistical analysis been performed appropriately and rigorously? 

Reviewer #1: Yes

Reviewer #2: Yes

3. Have the authors made all data underlying the findings in their manuscript fully available?

Reviewer #1: No

Reviewer #2: No

4. Is the manuscript presented in an intelligible fashion and written in standard English?

Reviewer #1: Yes

Reviewer #2: Yes

5. Review Comments to the Author

Reviewer #1: The study aimed to assess the association of haemoglobin (Hb) in the first 20 weeks of pregnancy, and restoration of low Hb levels, with pregnancy outcomes in Australia.

The study included 31,906 singletons births data obtained from 2011 to 2015 from hospital records. Women with both low and high Hb in the first 20 weeks of pregnancy had higher risks of adverse outcomes than those with normal Hb. Restoring Hb after 20 weeks did not improve most adverse outcome rates but did reduce risk of transfusion.

This is an important findings since suggest that women anemic in the first half of pregnancy probably were anemics since before pregnancy. This is suggested form one of the results of the study. This may allow to give treatment only to those women qualified as anemic before they get pregnant.

It is interesting the U shape for fetal outcomes. A previous paper demonstrated this U shape in populations living at low and at high altitude (Gonzales et al., 2009) indicating that association could be global.

The absence of improvement after restored Hb at the second half of pregnancy is also important finding suggesting that use of iron supplementation on this period of gestation should be limited. This is in accordance with a previous findings that increase in Hb concentration in a second booking after a normal Hb at first booking resulted in double the risk of small for gestational age (Gonzales et al., 2012).

In realtion to pregnant women with high Hb levels, authors suggest inadequate plasma volume expansion (hemoconcentration) or impaired response to inflammation and infection. Authors should also consider that women may also have high Hb levels before pregnancy.

Minor comments:

In the Table there is no number of cases for 140+ g/L group. Instead appears a column with percentages. Please check

Gonzales GF, Tapia V, Fort AL. Maternal and perinatal outcomes in second hemoglobin measurement in nonanemic women at first booking: effect of altitude of residence in peru. ISRN Obstet Gynecol. 2012;2012:368571

Gonzales GF, Steenland K, Tapia V. Maternal hemoglobin level and fetal outcome at low and high altitudes. Am J Physiol Regul Integr Comp Physiol. 2009 Nov;297(5):R1477-85. doi: 10.1152/ajpregu.00275.2009. Epub 2009 Sep 9.

Reviewer #2: Randall et al conducted a retrospective cohort study analyzing the association of haemoglobin values in early pregnancy (<20 weeks) with pregnancy outcomes, maternal post-partum haemorrhage and transfusion up to 6 weeks post-partum. Data was collected from two large public hospitals in New South Wales, Australia. An appropriately large group of subjects was included in the analysis - 31,906 singleton pregnancies - and the cohort was split into three groups based on their Hb levels (Hb <110g/L, 110-139g/L and >140g/L) for analysis. Overall, there was a U-shaped association between maternal Hb values prior to 20 weeks gestation and adverse pregnancy outcomes, with the Hb <110g/L group having the greatest percentage of adverse outcomes. Transfusion was the only outcome that displayed a linear relationship with decreased risk with increasing Hb levels. This study supports previous studies demonstrating that low maternal Hb along with high maternal hemoglobin results in increased negative pregnancy outcomes. Importantly, the study found that restoring the Hb during the pregnancy did not improve adverse outcomes except for the risk of postpartum transfusion. This study highlights the need to for further studies to understand why and how low and high Hb are associated with poorer outcomes and how these poor outcomes can be prevented before or during the pregnancy.

Questions/Comments:

1) Provide explanation for why lowest Hb was selected rather than the most recent Hb at <20 weeks of gestation.

2) It may be helpful to include the range of Hb values for both the low and high Hgb group.

3) Table 1. n not provided for >140g/L group, instead average percentage is presented in table

4) Could the authors speculate on why restoring Hb during the pregnancy did not improve most of the adverse outcomes?

6. PLOS authors have the option to publish the peer review history of their article (what does this mean?). If published, this will include your full peer review and any attached files.

Reviewer #1: No

Reviewer #2: No

---

## [Author Response · Author response to Decision Letter 0]

17 Oct 2019

Please see the uploaded Word file with a detailed response to the editor and reviewer comments.

---

## [Editor Report · Decision Letter 1]

30 Oct 2019

The association between haemoglobin levels in the first 20 weeks of pregnancy and pregnancy outcomes

PONE-D-19-19488R1

Dear Dr. Deborah Anne Randall,

We are pleased to inform you that your manuscript has been judged scientifically suitable for publication and will be formally accepted for publication once it complies with all outstanding technical requirements.

With kind regards,

Massimo Ciccozzi

Academic Editor

PLOS ONE
---

## [Editor Report · Acceptance letter]

6 Nov 2019

PONE-D-19-19488R1 

The association between haemoglobin levels in the first 20 weeks of pregnancy and pregnancy outcomes 

Dear Dr. Randall:

I am pleased to inform you that your manuscript has been deemed suitable for publication in PLOS ONE. Congratulations! Your manuscript is now with our production department. 

With kind regards,

on behalf of

Prof Massimo Ciccozzi 

Academic Editor

PLOS ONE